# CONTINUAL LEARNING IN RECURRENT NEURAL NETWORKS

**Benjamin Ehret\*, Christian Henning\*, Maria R. Cervera\*, Alexander Meulemans,
Johannes von Oswald, Benjamin F. Grewe**
\*Equal contribution

Institute of Neuroinformatics
University of Zürich and ETH Zürich
Zürich, Switzerland
`{behret,henningc,mariacer,ameulema,voswaldj,bgrewe}@ethz.ch`

## ABSTRACT

While a diverse collection of continual learning (CL) methods has been proposed to prevent catastrophic forgetting, a thorough investigation of their effectiveness for processing sequential data with recurrent neural networks (RNNs) is lacking. Here, we provide the first comprehensive evaluation of established CL methods on a variety of sequential data benchmarks. Specifically, we shed light on the particularities that arise when applying weight-importance methods, such as elastic weight consolidation, to RNNs. In contrast to feedforward networks, RNNs iteratively reuse a shared set of weights and require working memory to process input samples. We show that the performance of weight-importance methods is not directly affected by the length of the processed sequences, but rather by high working memory requirements, which lead to an increased need for stability at the cost of decreased plasticity for learning subsequent tasks. We additionally provide theoretical arguments supporting this interpretation by studying linear RNNs. Our study shows that established CL methods can be successfully ported to the recurrent case, and that a recent regularization approach based on hypernetworks outperforms weight-importance methods, thus emerging as a promising candidate for CL in RNNs. Overall, we provide insights on the differences between CL in feedforward networks and RNNs, while guiding towards effective solutions to tackle CL on sequential data.

## 1 INTRODUCTION

The ability to continually learn from a non-stationary data distribution while transferring and protecting past knowledge is known as continual learning (CL). This ability requires neural networks to be stable to prevent forgetting, but also plastic to learn novel information, which is referred to as the stability-plasticity dilemma (Grossberg, 2007; Mermillod et al., 2013). To address this dilemma, a variety of methods which tackle CL for static data with feedforward networks have been proposed (for reviews refer to Parisi et al. (2019) and van de Ven and Tolias (2019)). However, CL for sequential data has only received little attention, despite recent work confirming that recurrent neural networks (RNNs) also suffer from catastrophic forgetting (Schak and Gepperth, 2019).

A set of methods that holds great promise to address this problem are regularization methods, which work by constraining the update of certain parameters. These methods can be considered more versatile than competing approaches, since they do not require rehearsal of past data, nor an increase in model capacity, but can benefit from either of the two (e.g., Nguyen et al., 2018; Yoon et al., 2018). This makes regularization methods applicable to a broader variety of situations, e.g. when issues related to data privacy, storage, or limited computational resources during inference might arise. The most well-known regularization methods are *weight-importance methods*, such as elastic weight consolidation (EWC, Kirkpatrick et al. (2017a)) and synaptic intelligence (SI, Zenke et al. (2017)), which are based on assigning importance values to weights. Some of these have a direct probabilistic interpretation as prior-focused CL methods (Farquhar and Gal, 2018), for which

solutions of upcoming tasks must lie in the posterior parameter distribution of the current task (cf. Fig. 1b), highlighting the stability-plasticity dilemma. Whether this dilemma differently affects feedforward networks and RNNs, and whether weight-importance based methods can be used off the shelf for sequential data has remained unclear.

Here, we contribute to the development of CL approaches for sequential data in several ways.

- We provide a first comprehensive comparison of CL methods applied to sequential data. For this, we port a set of established CL methods for feedforward networks to RNNs and assess their performance thoroughly and fairly in a variety of settings.

- We identify elements that critically affect the stability-plasticity dilemma of weight-importance methods in RNNs. We empirically show that high requirements for working memory, i.e. the need to store and manipulate information when processing individual samples, lead to a saturation of weight importance values, making the RNN rigid and hindering its potential to learn new tasks. In contrast, this trade-off is not directly affected by the sheer recurrent reuse of the weights, related to the length of processed sequences. We complement these observations with a theoretical analysis of linear RNNs.

- We show that existing CL approaches can constitute strong baselines when compared in a standardized setting and if equivalent hyperparameter-optimization resources are granted. Moreover, we show that a CL regularization approach based on hypernetworks (von Oswald et al., 2020) mitigates the limitations of weight-importance methods in RNNs.

- We provide a code base[1] comprising all assessed methods as well as variants of four well known sequential datasets adapted to CL: the Copy Task (Graves et al., 2014), Sequential Stroke MNIST (Gulcehre et al., 2017), AudioSet (Gemmeke et al., 2017) and multilingual Part-of-Speech tagging (Nivre et al., 2016).

Taken together, our experimental and theoretical results facilitate the development of CL methods that are suited for sequential data.

## 2 RELATED WORK

**Continual learning with sequential data.** As in Parisi et al. (2019), we categorize CL methods for RNNs into regularization approaches, dynamic architectures and complementary memory systems.

Regularization approaches set optimization constraints on the update of certain network parameters without requiring a model of past input data. EWC, for example, uses weight importance values to limit further updates of weights that are considered essential for solving previous tasks (Kirkpatrick et al., 2017b). Throughout this work, we utilize a more mathematically sound and less memory-intensive version of this algorithm, called Online EWC (Huszár, 2018; Schwarz et al., 2018). Although a highly popular approach in feedforward networks, it has remained unclear how suitable EWC is in the context of sequential processing. Indeed, some studies report promising results in the context of natural language processing (NLP) (Madasu and Rao, 2020; Thompson et al., 2019), while others find that it performs poorly (Asghar et al., 2020; Cossu et al., 2020a; Li et al., 2020). Here, we conduct the first thorough investigation of EWC's performance on RNNs, and find that it can often be a suitable choice. A related CL approach that also relies on weight importance values is SI (Zenke et al., 2017). Variants of SI have been used for different sequential datasets, but have not been systematically compared against other established methods (Yang et al., 2019; Masse et al., 2018; Lee, 2017). Fixed expansion layers (Coop and Arel, 2012) are another method to limit the plasticity of weights and prevent forgetting, and in RNNs take the form of a sparsely activated layer between consecutive hidden states (Coop and Arel, 2013). Lastly, some regularization approaches rely on the use of non-overlapping and orthogonal representations to overcome catastrophic forgetting (French, 1992; 1994; 1970). Masse et al. (2018), for example, proposed the use of context-dependent random subnetworks, where weight changes are regularized by limiting plasticity to task-specific subnetworks. This eliminates forgetting for disjoint networks but leads to a reduction of available capacity per task. In concurrent work, Duncker et al. (2020) introduced a learning rule which aims to optimize the use of the activity-defined subspace in RNNs learning multiple tasks. When tasks are different,

---

[1]Source code for all experiments (including all baselines) is available at `https://github.com/mariacer/cl_in_rnns`.

catastrophic interference is avoided by forcing the use of task-specific orthogonal subspaces, whereas the reuse of dynamics is encouraged across tasks that are similar.

Dynamic architecture approaches, which rely on the addition of neural resources to mitigate catastrophic forgetting, have also been applied to RNNs. Cossu et al. (2020a) presented a combination of progressive networks (Rusu et al., 2016) and gating autoencoders (Aljundi et al., 2017), where an RNN module is added for each new task and the reconstruction error of task-specific autoencoders is used to infer the RNN module to be used. Arguably, the main limitation of this type of approach is the increase in the number of parameters with the number of tasks, although methods have been presented that add resources for each new task only if needed (Tsuda et al., 2020).

Finally, complementary memory systems have also been applied to the retention of sequential information. In an early work, Ans et al. (2004) proposed a secondary network that generates patterns for rehearsing previously learned information. Asghar et al. (2020) suggested using an external memory that is progressively increased when new information is encountered. Sodhani et al. (2020) combined an external memory with Net2Net (Chen et al., 2016), such that the network capacity can be extended while maintaining memories. The major drawback of complementary memory systems is that they either violate CL desiderata by storing past data, or rely on the ability to learn a generative model, a task that arguably scales poorly to complex data. We discuss related work in a broader context in supplementary materials (SM D).

**Hypernetworks.** Introduced by Ha et al. (2017), the term *hypernetwork* refers to a neural network that generates the weights of another network. The idea can be traced back to Schmidhuber (1992), who already suggested that a recurrent hypernetwork could be used for learning to learn (Schmidhuber, 1993). Importantly, hypernetworks can make use of the fact that parameters in a neural network possess compressible structure (Denil et al., 2013; Han et al., 2015). Indeed, Ha et al. (2017) showed that the number of trainable weights of feed-forward architectures can be reduced via hypernetworks. More recently, hypernetworks have been adapted for CL (He et al., 2019; von Oswald et al., 2020), but not for learning with sequential data.

## 3 METHODS

**Recurrent Neural Networks.** We consider discrete-time RNNs. At timestep $t$, the network's output $\hat{\mathbf{y}}_t$ and hidden state $\mathbf{h}_t$ are given by $(\hat{\mathbf{y}}_t, \mathbf{h}_t) = f_{\text{step}}(\mathbf{x}_t, \mathbf{h}_{t-1}, \psi)$, where $\mathbf{x}_t$ denotes the input at time $t$ and $\psi$ the parameters of the network (Cho et al., 2014; Elman, 1990; Hochreiter and Schmidhuber, 1997a). In this work, we consider either vanilla RNNs (based on Elman (1990)), LSTMs (Hochreiter and Schmidhuber, 1997a) or BiLSTMs (Schuster and Paliwal, 1997).

**Naive baselines.** We consider the following naive baselines. **Fine-tuning** refers to training an RNN sequentially on all tasks without any CL protection. Each task has a different output head (multi-head), and the heads of previously learned tasks are kept fixed. **Multitask** describes the parallel training on all tasks (no CL). To keep approaches comparable, the multitask baseline uses a multi-head output. Because we focus on methods with a comparable number of parameters, we summarize approaches that allocate a different model per task in the **From-scratch** baseline, where a different model is trained separately for each task, noting that performance improvements are likely to arise in related methods (such as Cossu et al. (2020a)) whenever knowledge transfer is possible.

**Continual learning baselines.** We consider a diverse set of established CL methods and investigate their performance in RNNs. **Online EWC** (Huszár, 2018; Kirkpatrick et al., 2017a; Schwarz et al., 2018) and **SI** (Zenke et al., 2017) are different weight-importance CL methods. A simple weighted L2 regularization ensures that the neural network is more rigid in weight directions that are considered *important* for previous tasks, i.e., the loss for the $K$-th task is given by

$$\mathcal{L}(\psi, \mathcal{D}_K) = \mathcal{L}_{\text{task}}(\psi, \mathcal{D}_K) + \lambda \sum_{i=1}^{|\psi|} \omega_i (\psi_i - \tilde{\psi}_i^{(K-1)})^2 \tag{1}$$

where $\lambda$ is the regularization strength, $\omega_i$ is the *importance* associated with $\psi_i$ (cf. SM B.5 and B.6) and $\tilde{\psi}^{(K-1)}$ denotes the main network weights $\psi$ that were checkpointed after learning task $K - 1$. We denote by **HNET** a different regularization approach based on hypernetworks that was recently proposed by von Oswald et al. (2020). A hypernetwork (Ha et al., 2017) is a neural network $\psi = h(\mathbf{e}, \theta)$ with parameters $\theta$ and input embeddings $\mathbf{e}$ that generates the weights of a

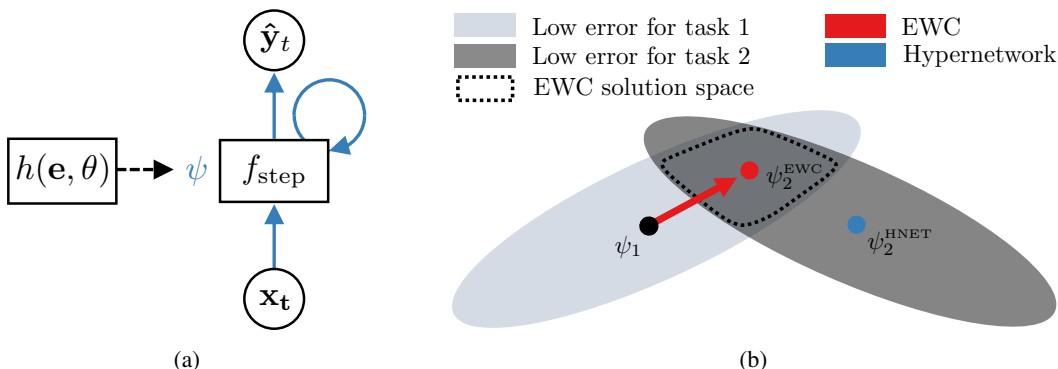

Figure 1: **(a)**: A hypernetwork $h(\mathbf{e}, \theta)$ produces the weights $\psi$ of a recurrent main network $f$ conditioned on $\mathbf{e}$. **(b)**: Here, we illustrate a hypernetwork-based CL approach versus a weight-importance method (such as EWC). Both methods start learning a second task from a common solution $\psi_1$. EWC is rigid along certain directions in weight space, which leads to a trade-off solution $\psi_2^{\text{EWC}}$ when seeking good optima for the upcoming task (cf. Farquhar and Gal (2018)). The hypernetwork-based approach (Eq. 2) has no such trade-off built into its objective; it is only limited by the optimization algorithm and network capacity, and is capable to output the task-specific solutions $\psi_1$ and $\psi_2^{\text{HNET}}$ (figure adapted from Kirkpatrick et al. (2017a)).

main network. This method sidesteps the problem of finding a compromise between tasks with a shared model $\psi$, by generating a task-specific model $\psi^{(k)}$ from a low-dimensional embedding space via a shared hypernetwork in which the weights $\theta$ and embeddings $\mathbf{e}$ are continually learned. In contrast to von Oswald et al. (2020), we focus here on RNNs as main networks: $f_{\text{step}}(\mathbf{x}_t, \mathbf{h}_{t-1}, \psi) = f_{\text{step}}(\mathbf{x}_t, \mathbf{h}_{t-1}, h(\mathbf{e}, \theta))$ (Fig. 1a). Crucially, this method has the advantage of not being noticeably affected by the recurrent nature of the main network, since CL is delegated to a feedforward meta-model, where forgetting is avoided based on a simple L2-regularization of its output. For a fair comparison, we ensure that the number of trainable parameters is comparable to other baselines by focusing on chunked hypernetworks (von Oswald et al., 2020), and enforcing: $\left| \theta \cup \{\mathbf{e}_k\}_{k=1}^{K} \right| \leq |\psi|$. Further details can be found in SM B.4. **Masking** (or *context-dependent gating*, Masse et al. (2018)) applies a binary random mask per task for all hidden units of a multi-head network, and can be seen as a simple method for selecting a different subnetwork per task. Since catastrophic interference can occur because of the overlap between subnetworks, this method can be combined with other CL methods such as SI (**Masking+SI**). We also consider methods based on replaying input data from previous tasks, either via a sequentially trained generative model (Shin et al., 2017; van de Ven and Tolias, 2018), denoted **Generative Replay**, or by maintaining a small subset of previous training data (Rebuffi et al., 2017; Nguyen et al., 2018), denoted **Coresets-**$N$, where $N$ refers to the number of samples stored for each task. Target outputs for replayed data are obtained via a copy of the main network, stored before training on the current task (detailed baseline descriptions in SM B).

**Task Identity.** We assume that task identity is provided to the system during training and inference, either by selecting the correct output head or by feeding the correct task embedding $\mathbf{e}_k$ into the hypernetwork, and elaborate in SM G.12 on how to overcome this limitation.

## 4   Analysis of weight-importance methods

Weight-importance methods have widely been used in feedforward networks, but whether or not they are suited for tackling CL in RNNs has remained unclear. Here, we investigate the particularities that weight-importance methods face when applied to RNNs. As opposed to feedforward networks, RNNs provide a natural way to process temporal sequences. Because their hidden states are a function of newly incoming inputs as well as their own activity in the previous timestep, RNNs are able to store and manipulate sample-specific information within their hidden activity, thus providing a form of working memory. Importantly, processing input sequences one sequence element at a time results in the reuse of recurrent weights for a number of times that is equal to the length of the input sequence. In this section, we investigate whether weight importance values are directly affected by working

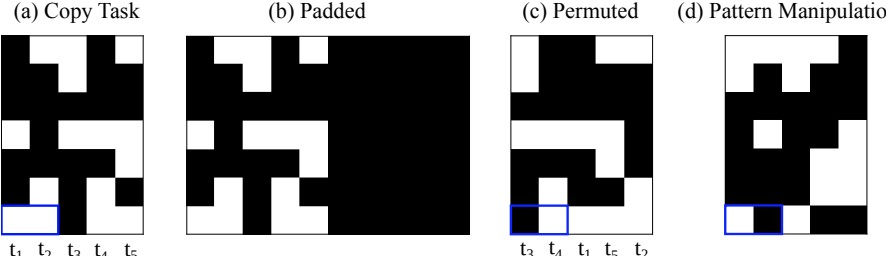

Figure 2: Variants of the Copy Task. **(a)**: Example pattern of the basic Copy Task where input and output patterns are identical. **(b)**: In Padded Copy Task, the input is padded with zeros, but the output consists only of the pattern itself, i.e. to (a). **(c)**: In Permuted Copy, the output is a time-permuted version of the presented input in (a). **(d)**: In the Pattern Manipulation Task ($r = 1$), the output corresponds to an XOR operation between the original input (a) and a permuted version of the input (c). The blue rectangle highlights this operation for two specific bits.

memory requirements or by the length of processed sequences. For this, we develop some intuitions by studying linear RNNs, which we then test in non-linear RNNs using a synthetic dataset.

First, we obtain some theoretical insights by analysing how linear RNNs can learn to solve a set of tasks (for details refer to SM C). Whenever task-specific output heads are not rich enough to model task variabilities, RNNs trained with methods whose recurrent computation is not task-conditioned (e.g. weight-importance methods) must solve all tasks simultaneously within their hidden space. In an extreme scenario where tasks are so different that they cannot share any useful computation, it becomes clear that task interferences can be prevented if the information relevant to each task resides in task-specific orthogonal subspaces (refer to SM G.10 for a discussion of scenarios with task similarity). Maintaining this structure within the hidden space imposes certain constraints on the recurrent weights. Our theory shows these constraints increase with the number of tasks and with the dimensionality of the task-specific subspaces. Based on these theoretical insights, we hypothesize that also in nonlinear RNNs increasing working memory requirements cause high weight rigidity (as illustrated by high importance values), whereas the recurrent reuse of weights is not a driving factor.

To test this hypothesis, we explore a synthetic dataset consisting of several variations of the Copy Task (Graves et al., 2014), in which a random binary input pattern has to be recalled by the network after a stop bit is observed (see Fig. 2, and SM E.1 for details). For all Copy Task experiments, we use vanilla RNNs combined with orthogonal regularization (see SM G.2). We denote the length (number of timesteps) of the binary input pattern to be copied by $p$, and the actual number of timesteps until the stop bit by $i$ (examples can be found in SM Fig. S1). This distinction allows us to consider two variants, the basic Copy Task where $p = i$, and the *Padded Copy Task* where $i > p$ (Fig. 2 a and b). In this variant, we zero-pad a binary input pattern of length $p$ for $i - p$ timesteps until the occurrence of the stop bit, resulting in an input sequence with $i$ timesteps. Specifically, we consider a set of Copy Tasks[2] with varying input lengths $i$ and, either a fixed pattern length $p = 5$, or a pattern length tied to the input length ($p = i$). This allows us to disentangle how sequence length and memory load affect weight importance.

As in `Online EWC`, we calculate weight importance as the diagonal elements of the empirical Fisher information matrix (see SM B.5). To quantify memory load, we study the intrinsic dimensionality of the hidden state of the RNN using principal component analysis (PCA), once networks have been trained to achieve near optimal performance (above 99%). We define the intrinsic dimensionality as the number of principal components that are needed to explain 75% of the variance.

As expected, the intrinsic dimensionality of the hidden space increases during input pattern presentation and peaks after $p$ timesteps, i.e. at the stop bit for tasks with $p = i$ (Fig. 3a) and $i - p$ timesteps before the stop bit if $p = 5$ remains fixed (Fig. 3b).[3] Weight importance values rapidly increase with memory requirements ($p$), but not sequence length (increasing $i$, fixed $p$) (Fig. 3c). The same trend is

---

[2]Note, these tasks are learned independently to isolate the effects of $i$ and $p$ on weight importance values.

[3]Note, Fig. 3a shows a decreased dimensionality at $t = 0$ for $i = 35$ or $i = 40$ compared to $i = 30$. We hypothesize that this is due to a need to non-linearly encode information into the hidden state for large $i$, and verified that the dimensionality increases with $i$ when using Kernel PCA (Schölkopf et al., 1997), data not shown.

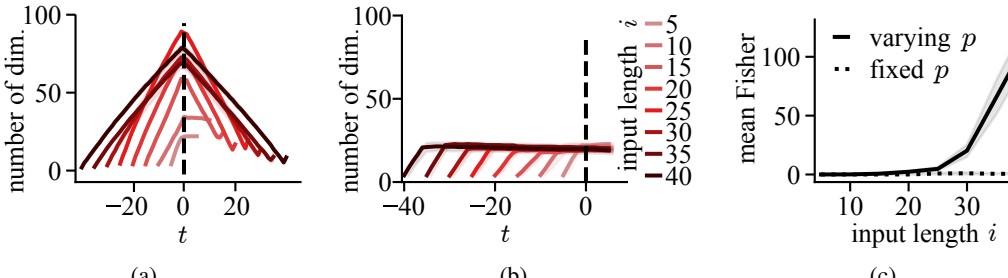

Figure 3: **(a)** Intrinsic dimensionality per timestep of the 256-dimensional RNN hidden space $\mathbf{h}_t$ for the basic Copy Task, where input and pattern lengths are tied ($i = p$). The stop bit (dotted black line) is shown at time $t = 0$ (Mean $\pm$ SD, $n = 5$). **(b)** Same as (a) for the Padded Copy Task, where the pattern length is fixed ($p = 5$) but input length $i$ varies. In (a) and (b), dimensionality of the hidden state space increases only during input pattern presentation. **(c)** Mean Fisher values (weight-importance values in `Online EWC`) of recurrent weights after learning the Copy Task (solid line, $p = 5, 10, ...40$) or the Padded Copy Task (dotted line, $p = 5$) independently for an increasing set of sequence lengths $i$ (Mean $\pm$ SD, $n = 5$).

observed when computing weight-importance values according to `SI` (cf. SM G.3), and when using LSTMs instead of vanilla RNNs (cf. SM G.4). We extend this analysis to a CL setting in SM G.5.

Overall, this analysis reveals that weight-importance methods are affected by the processing and storage required by the task, but not by the sequential nature of the data, even if the same set of weights is reused over many timesteps. This could cause weight-importance methods to suffer from a saturation of importance values when working memory requirements are high, which could in turn reduce their plasticity for learning new tasks. We explore this in the next section.

## 5 CONTINUAL LEARNING EXPERIMENTS

To highlight strengths and weaknesses of different CL methods in various settings, we performed experiments on one synthetic and two real-world sequential datasets, using different types of RNNs. We distinguish between `during` and `final` accuracies. The `during` accuracy of a CL experiment is obtained by taking the mean over the test accuracy from each task right after it has been trained on, i.e., when tasks have not yet been subject to forgetting. The `final` accuracy describes the mean test accuracy over all tasks obtained after the last task has been learned.

For all reported methods, results were obtained via an extensive hyperparameter search, where the hyperparameter configuration of the run with best `final` accuracy was selected and subsequently tested on multiple random seeds (experimental details in SM F). We provide additional CL experiments on multilingual NLP data in SM G.9.

### 5.1 VARIATIONS OF THE COPY TASK

After exposing the challenges that weight-importance methods face when dynamically processing data, we explore how these manifest in a CL scenario. We compare weight-importance methods against other CL approaches, with a particular focus on `HNET`, which can in principle bypass those challenges. We transform the Copy Task into a set of CL tasks by applying for each task $k$ a different random time-permutation $\pi^{(k)}$. In this setting, which we denote *Permuted Copy Task*, each timestep $t_i \in \{1, \ldots, p\}$ from the input pattern has to be recalled at output timestep $t_o = \pi^{(k)}(t_i)$ (Fig. 2c). We perform these experiments on vanilla RNNs. First, we evaluate all methods in a relatively simple scenario with five tasks using $p = i = 5$ (Table 1). `Online EWC` achieves very high performance, and `HNET` reaches close to 100% accuracy. The random subnetworks in `Masking` can learn individual tasks to perfection. However, weight changes within subnetworks, which result from random overlaps, cause severe performance drops and show the need to add stabilization mechanisms, e.g., `Masking+SI`. Since the input data distribution is relatively simple and identical across tasks, learning a generative model is feasible, which is illustrated by the performance of `Gen. Replay`.

Table 1: Mean `during` and `final` accuracies for the Permuted Copy Task with $p = i = 5$ using 5 tasks each (Mean $\pm$ SEM in %, $n = 10$).

|  | **during** | **final** |
|---|---|---|
| Multitask | N/A | $99.87 \pm 0.05$ |
| From-scratch | N/A | $100.00 \pm 0.00$ |
| Fine-tuning | $99.99 \pm 0.00$ | $71.05 \pm 0.13$ |
| HNET | $99.98 \pm 0.00$ | $99.96 \pm 0.01$ |
| Online EWC | $99.93 \pm 0.01$ | $98.66 \pm 0.14$ |
| SI | $98.41 \pm 0.06$ | $94.03 \pm 0.24$ |
| Masking | $99.53 \pm 0.26$ | $72.31 \pm 0.82$ |
| Masking+SI | $99.40 \pm 0.25$ | $99.40 \pm 0.25$ |
| Gen. Replay | $100.00 \pm 0.00$ | $100.00 \pm 0.00$ |
| Coresets-100 | $100.00 \pm 0.00$ | $99.94 \pm 0.00$ |

Table 2: Detailed `Online EWC/HNET` comparisons (Mean $\pm$ SEM in %, $n = 5$). $r$ is the number of task-specific random permutations.

|  | **during** | **final** |
|---|---|---|
| **Padded Copy Task** | | |
| HNET | $100.00 \pm 0.00$ | $100.00 \pm 0.00$ |
| Online EWC | $97.94 \pm 0.09$ | $97.89 \pm 0.10$ |
| **Pattern Manipulation Task r $= 1$** | | |
| HNET | $100.00 \pm 0.00$ | $99.84 \pm 0.15$ |
| Online EWC | $98.52 \pm 0.27$ | $95.45 \pm 0.17$ |
| **Pattern Manipulation Task r $= 5$** | | |
| HNET | $95.73 \pm 1.44$ | $93.87 \pm 1.24$ |
| Online EWC | $87.40 \pm 4.53$ | $81.80 \pm 3.25$ |

In the following, we focus on a comparison between `Online EWC` and `HNET` to further investigate how these methods are affected by sequence length and working memory requirements. We first test whether `Online EWC` is affected by sequence length by investigating the Permuted Copy Task at $p = 5, i = 25$ using 5 tasks. As Table 2 shows, the performance of both methods is not markedly affected by sequence length. Interestingly, the results are slightly better for longer sequences with both methods, which can be due to an increased processing time between input presentation and recall. Next, we compare the performance of `Online EWC` and `HNET` in a set of tasks for which working memory requirements can be easily controlled. In this setting, referred to as *Pattern Manipulation Task*, difficulty is controlled by a set of $r$ task-specific random permutations along the time axis (Fig. 2d). The output is computed from the input pattern by applying a binary XOR operation iteratively with all of its $r$ permutations (i.e. the result of the XOR between the input and its first permutation will then undergo a second XOR operation with the second permutation of the input, and so on). Note that this variant substantially differs from previous Copy Task variations, since the processing of input patterns is now both input- and task-dependent. As shown in Table 2, `Online EWC` experiences a larger drop with increased task difficulty than `HNET`, confirming that it is more severely affected by working memory requirements. Finally, we investigate the difference between single-head and multi-head settings, as well as task conditional processing in SM G.13.

## 5.2 SEQUENTIAL STROKE MNIST

To test whether the results from the synthetic Copy Task hold true for real world data we turned to a sequential digit recognition task where task difficulty can be directly controlled. In the Stroke MNIST (SMNIST) dataset (de Jong, 2016), MNIST images (LeCun et al., 1998) are represented as sequences of pen displacements, that result in the original digits when drawn in order. We adapt this dataset to a CL scenario by splitting it into five binary classification problems (digits 0 vs 1, 2 vs 3, etc.), reminiscent of the popular Split-MNIST experiment commonly used to benchmark CL methods on static data (Zenke et al., 2017). Interestingly, this dataset allows exploring how the performance of different CL methods depends on the difficulty of individual tasks by generalizing the notion of Split-SMNIST to sequences of $m$ SMNIST samples (cf. Gulcehre et al., 2017), where each sequence contains only two types of digits (e.g. 2332 or 7767 for $m = 4$). To obtain a binary decision problem, we randomly group all $2^m$ possible sequences within a task into two classes. This ensures that despite increasing levels of task difficulty, as determined by $m$, chance level is not affected. Crucially, an increase in $m$ leads to an increase in the amount of information that needs to be stored and manipulated per input sequence, and therefore allows exploring the effect that increasing working memory requirements have on different CL methods in a real-world dataset.

We train LSTMs on five tasks for an increasing number of digits per sequence ($m = 1, 2, 3, 4$) and observe that methods are differently affected by the task difficulty level (see Fig. 4). For $m = 1$ `Online EWC`, `SI` and `HNET` all achieve above 97% performance. However for $m = 4$ the performance of `Online EWC` and `SI` drops to $73.16 \pm 0.98\%$ and $69.58 \pm 0.55\%$ respectively, while the hypernetwork approach successfully classifies $94.42 \pm 1.85\%$ of all inputs. Thus weight-importance methods seem more strongly affected than `HNET` by an increase in task complexity and working memory requirements. Interestingly, the performance gap depends on the experimental setup as outlined in SM G.13. `Coresets` perform slightly worse, especially when task-complexity increases. `Masking+SI`, which trades-off network capacity for the ability of finding solutions in a

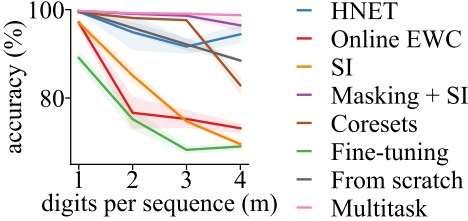

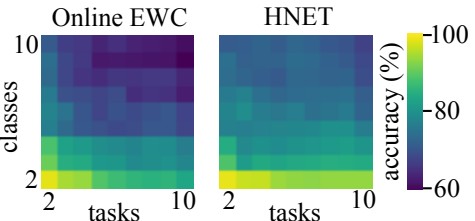

Figure 4: Mean `final` accuracies for four Split Sequential-SMNIST experiments, each comprising five tasks (Mean ± SEM, $n = 10$).

Figure 5: Mean `final` accuracies for Split-AudioSet experiments with varying number of classes and tasks.

less rigid subnetwork, emerges as the preferable method for this experiment. We additionally list `during` accuracies for all methods in SM Table 5 and discuss the use of replay for Split-SMNIST in SM G.7. Finally we show that, consistent with our Copy Task results, the performance of weight-importance methods is not significantly affected when sequence lengths are increased without a concomitant increase in working memory (cf. SM. G.8).

## 5.3 AUDIOSET

AudioSet (Gemmeke et al., 2017) is a dataset of manually annotated audio events. It consists of 10-second audio snippets which have been preprocessed by a VGG network to extract 128-dimensional feature vectors at 1 Hz. This dataset has been previously adapted for CL by Kemker and Kanan (2018) and Kemker et al. (2018), whose particular split has not been made public, and by Cossu et al. (2020b), for which the test set size largely differed across classes. We therefore created a new variant, which we call *Split-AudioSet-10*, containing 10 tasks with 10 classes each (see SM F.3 for details).

Table 3: Mean `during` and `final` accuracies for the Split-AudioSet-10 experiments (Mean ± SEM in %, $n = 10$).

|  | during | final |
|---|---|---|
| Multitask | N/A | 77.31 ± 0.10 |
| From-scratch | N/A | 79.06 ± 0.11 |
| Fine-tuning | 71.95 ± 0.24 | 49.02 ± 1.00 |
| HNET | 73.05 ± 0.45 | 71.76 ± 0.62 |
| Online EWC | 68.82 ± 0.20 | 65.56 ± 0.35 |
| SI | 67.66 ± 0.10 | 66.92 ± 0.04 |
| Masking | 75.81 ± 0.15 | 50.87 ± 1.09 |
| Masking+SI | 64.88 ± 0.19 | 64.86 ± 0.20 |
| Coresets-100 | 74.25 ± 0.11 | 72.30 ± 0.11 |
| Coresets-500 | 77.03 ± 0.08 | 73.90 ± 0.07 |

The results obtained in this dataset using LSTMs are listed in Table 3. `HNET` is the strongest among regularization based methods, and is only outperformed by `Coresets`, which rely on storing past data. `Masking` during accuracies indicate that random subnetworks have enough capacity to learn individual tasks, but low `final` accuracies suggest that catastrophic forgetting occurs, presumably because of the overlap between subnetworks. This is partly solved in `Masking+SI` by introducing stabilization which, however, reduces plasticity for learning new tasks. In contrast to the findings of Sec. 5.2 and SM G.9, `Masking+SI` performs worst among regularization approaches, indicating that trading-off capacity for complex datasets can be harmful. The `From-scratch` baseline outperforms other methods, which is explained by the fact that it trains a separate model per task, leading to 10 times more network capacity. Notably, we were not able to successfully train a `Generative Replay` model on this dataset despite extensive hyperparameter search. Together with the results in Sec. 5.1, this highlights that the performance of `Generative Replay` depends on the complexity of the input data distribution, and not necessarily on the CL nature of the problem. To further investigate the stability-plasticity trade-off, we tested `HNET` and `Online EWC` across a range of difficulty levels in individual tasks. This can be controlled by the number of classes to be learned within each task, which we varied from two to ten. For both methods we used the best hyperparameters found for Split-AudioSet-10. Since the performance of `Online EWC` strongly depends on the regularization parameter $\lambda$, we tuned this value to achieve optimal results in each setting (cf. Fig. S2). Fig. 5 shows the task-averaged `final` accuracies for the different task-difficulty settings. While `HNET` performance is primarily affected by task difficulty but not by the number of tasks, results for `Online EWC` show an interplay between task difficulty and the ability to retain good performance on many tasks. These results provide further evidence that the hypernetwork-based approach can resolve the limitations of weight-importance CL methods for sequential data.

## 6 DISCUSSION

**The stability-plasticity dilemma with sequential data.** Weight-importance methods address CL by progressively constraining a network's weights, directly trading plasticity for stability. In the case of RNNs, weights are subject to additional constraints, since the same set of weights is reused across time to dynamically process an input stream of data. We show that increased working memory requirements, resulting from more complex processing within individual tasks, lead to high weight-importance values and can hinder the ability to learn future tasks (cf. Sec. 4). On the contrary, we find that longer sequence lengths do not impact performance for a fixed level of task complexity (cf. Fig. 3b, Table 2), suggesting that weight reuse doesn't interfere with the RNN's ability to retain previous knowledge. These observations are consistent with our theoretical analysis of linear RNNs (SM C), which predicts that more challenging processing within individual tasks leads to increased interference between tasks. This aggravates the stability-plasticity dilemma in weight-importance based methods, which we confirm in a range of experiments.

**Benefits of a hypernetwork-based CL approach for sequential data.** We propose that a hypernetwork-based approach (von Oswald et al., 2020) can alleviate the stability-plasticity dilemma when continually learning with RNNs. Since stability is outsourced to a regularizer that does not directly limit the plasticity of main network weights (cf. SM Eq. 2), this approach has more flexibility than weight-importance methods for finding new solutions, as shown by our experiments. Although `Coresets` and `Generative Replay` perform better, these approaches might not always be applicable; `Coresets` rely on the storage of past data (which might not always be feasible for privacy or storage reasons), and `Generative Replay` scales poorly to complex data. On the contrary, an approach based on hypernetworks has the versatility of regularization approaches and can be used in a variety of situations.

**Future avenues for CL with RNNs.** As discussed in SM G.13, an interesting avenue for weight-importance methods is the use of task-conditional processing in order to overcome the need of solving all learned tasks in parallel. Although hypernetworks overcome this limitation by design, they introduce additional optimization challenges, especially in conjunction with vanilla RNNs (cf. SM G.2), which leaves room for future improvements. An interesting direction is the use of a recurrent hypernetwork to generate timestep-specific weights in the main RNN (Ha et al., 2017; Suarez, 2017). Although a naive application of this combination for CL using SM Eq. 2 would come at the cost of a linear increase in computation with the number of timesteps, this problem can be elegantly sidestepped by the use of a feed-forward hypernetwork that generates the weights of the recurrent hypernetwork. SM Eq. 2 can then simply be applied to the static output of this *hyper-hypernetwork*, protecting a set of timestep-specific weights per task without the need to increase the regularization budget.

## 7 CONCLUSION

Our work advances the CL community in three ways. First, by systematically evaluating the performance of established CL methods when applied to RNNs, we provide extensive baselines that can serve as reference for future studies on CL with sequential data. Second, we use theoretical arguments derived from linear RNNs to hypothesize limitations of weight-importance based CL in the context of recurrent computation, and provide empirical evidence to support these statements. Third, derived from these insights, we suggest that an approach based on hypernetworks mitigates the stability-plasticity dilemma, and show that it outperforms weight-importance methods on synthetic as well as real-world data. Finally, our work discusses several future improvements and directions of CL approaches for sequential data.

## ACKNOWLEDGEMENTS

This work was supported by the Swiss National Science Foundation (B.F.G. CRSII5-173721 and 315230_189251), ETH project funding (B.F.G. ETH-20 19-01) and funding from the Swiss Data Science Center (B.F.G, C17-18, J. v. O. P18-03). We especially thank João Sacramento for insightful discussions, ideas and feedback. We would also like to thank Nikola Nikolov and Seijin Kobayashi for helpful advice and James Runnalls for proofreading our manuscript.

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
