# OpenReview forum: "Continual learning in recurrent neural networks"
_ICLR.cc/2021/Conference — ICLR 2021 Poster_

### Official Review · AnonReviewer4 · 2020-10-27
**This review on continual learning for RNNs provides a very valuable service to the community.**

**Rating:** 7
**Confidence:** 4

**Review:**

Pros: Most work on continual learning addresses only feedforward networks. This paper
provides apparently the first systematic discussion and comparison of CL methods for RNNs
Thereby it provides an important service to the community. The material is presented thoroughly in the Suppl.

Cons:
Provides on the other hand less algorithmic innovation. In particular, it focuses on methods related to (Oswald et al., 2020), a paper that was accidentally omitted from the reference list, but apparently is this ICLR 2020 paper that contains related material:
von Oswald, J., Henning, C., Sacramento, J., & Grewe, B. F. (2019). Continual learning with hypernetworks. arXiv preprint arXiv:1906.00695. ICLR 2020.

I am uncertain about the generalizability of results that were demonstrated for the chosen benchmark tasks. In particular, the conceptually important distinction between challenges arising from working memory load and sequence length is tested by variations of the copy task with padded inputs, where relevant and irrelevant input bits are distinguished in a very simple way that is hardly met by real-world scenarios.

There may also be differences arising from different types of RNNs, and it is not clear to me to what extent one can make conclusions about all of them by testing on just one type.

I tend to vote for accept.

---

> ### Author Response · Authors · 2020-11-20
> **Reply to AnonReviewer4**
>
> We thank you for the constructive feedback and for the appreciation of our work. We have tried to address all your concerns, which we outline below.
>
>   - *Provides on the other hand less algorithmic innovation. In particular, it focuses on methods related to (Oswald et al., 2020), a paper that was accidentally omitted from the reference list, but apparently is this ICLR 2020 paper that contains related material: von Oswald, J., Henning, C., Sacramento, J., & Grewe, B. F. (2019). Continual learning with hypernetworks. arXiv preprint arXiv:1906.00695. ICLR 2020.*
>
> We may misunderstand the raised concern, but the paper of von Oswald et al. which was first published on arXiv in 2019 is identical to the version that was peer-reviewed and accepted at ICLR in 2020. Therefore we only cited the ICLR version.
>
>   - *I am uncertain about the generalizability of results that were demonstrated for the chosen benchmark tasks. In particular, the conceptually important distinction between challenges arising from working memory load and sequence length is tested by variations of the copy task with padded inputs, where relevant and irrelevant input bits are distinguished in a very simple way that is hardly met by real-world scenarios.*
>
> We thank AnonReviewer4 for this comment. Indeed, these two factors can be independently controlled in the Copy Task, but they are often entangled in real-world scenarios. To verify whether our observations on this synthetic task hold for more complex scenarios, we explored three real-world datasets and linked these to our original analysis whenever possible. In particular, the SSMNIST experiments confirm that an increase in working memory requirements (due to an increase in the number of digits per task) correlates with a significant drop in performance for weight-importance methods, but not for the hypernetwork approach. However, because increasing the number of digits per task also leads to an increase in the weight reuse, it is indeed not clear whether weight reuse is not the factor affecting weight-importance methods. To control for this, we now complement our analyses with an additional SSMNIST experiment, where sequence length is increased without a concomitant increase in working memory (SM G.8). Rather than using zero-padding, we achieve this by upsampling the original stroke sequences, i.e. by increasing temporal resolution and thereby adding redundant information. Consistent with our interpretation of the Copy Task results, these new experiments confirm for a real-world task that a sole increase in the sequence length doesn’t lead to a drop in performance of weight-importance methods.
>
>   - *There may also be differences arising from different types of RNNs, and it is not clear to me to what extent one can make conclusions about all of them by testing on just one type.*
>
> We understand the concern raised by AnonReviewer4 regarding the generalizability of our results to other types of RNNs. For this reason, we repeated the Copy Task analyses usingLSTMs instead of vanilla RNNs (SM G.4). These analyses show that, also in LSTMs, i) the dimensionality of the hidden space increases with higher working memory requirements (increasing pattern length) but not with a mere increase in weight reuse, and that ii) this increase correlates with an increase in importance values. This is consistent with the results obtained for vanilla RNNs, and highlights the generality of our results across RNN architectures.

---

### Official Review · AnonReviewer2 · 2020-10-28
**Extensive study and convincing results**

**Rating:** 6
**Confidence:** 3

**Review:**

This paper provides a systematic evaluation of the performance of different CL methods on RNN. The study suggests that high working memory requirements increase difficulty of learning new tasks, while the average length of input sequence is not strictly related to the difficulty of learning new tasks. The author proposes to overcome this problem by using a hypernetwork-based CL approach, which shows promising results in the experiments.

Strength:
* The paper provides extensive study to compare different continual learning methods.
* The conclusion is well supported by analysis of intrinsic dimension and performance on different tasks.
* The paper is well written and easy to follow

Weaknesses
* It would be interesting to see results on more realistic tasks, like sentence classification.
* The conclusion on working memory requirement didn’t consider the possibility of knowledge sharing between tasks. For example, two complicated tasks may share a common sub-network that is essential for solving both tasks. Such that, in an ideal situation, the model doesn't need to allocate large amounts of extra resources to learn the second task. It would be interesting to see how different CL methods can reuse knowledge learnt from previous tasks.

---

> ### Author Response · Authors · 2020-11-20
> **Reply to AnonReviewer2**
>
> We thank you for the careful review and the overall positive feedback. Below, we reply to all the raised concerns point-by-point.
>
>   - *It would be interesting to see results on more realistic tasks, like sentence classification.*
>
> We agree that it would be interesting to show results on a wider diversity of problems related to sequential processing. However, because a thorough investigation of all considered methods requires immense computational resources, we were forced to carefully select which datasets to consider. The three real-world datasets we explored were selected to ensure diversity in the type of tasks (i.e. classifying entire sequences as in Audioset or SSMNIST vs. assigning one label per timestep as in PoS tagging) and in the input domains (i.e. sound in Audioset, images in SSMNIST and text in PoS tagging). Therefore, our results include both a realistic NLP setting (the Part-of-Speech tagging task presented in SM G.9)  where a single model needs to learn to tag sentences from a different language in each task, and a realistic sequence classification task (Audioset), performed on audio snippets rather than text. As the general conclusions from those experiments coincide, they are likely to be transferable to related scenarios such as sentence classification.
>
>   - *The conclusion on working memory requirement didn’t consider the possibility of knowledge sharing between tasks. For example, two complicated tasks may share a common sub-network that is essential for solving both tasks. Such that, in an ideal situation, the model doesn't need to allocate large amounts of extra resources to learn the second task. It would be interesting to see how different CL methods can reuse knowledge learnt from previous tasks.*
>
> We thank AnonReviewer2 for this interesting comment. Indeed, the scenario we consider for the theoretical analysis is an extreme case and, most commonly, real-world tasks benefit from some form of knowledge sharing. In fact, in linear RNNs, it is easy to see that whenever the subspaces associated with individual tasks may overlap, the overall dimensionality of the used hidden space can be less than the sum of dimensionalities of individual task-related subspaces, thus freeing up capacity in the recurrent weights to learn new tasks. In other words, together with the working memory of individual tasks and the number of tasks, task similarity will also play a role in the effectiveness of weight-importance methods for RNNs. To clarify this point, we added a comment to the linear RNN analysis section in SM C. Furthermore, inspired by this comment, we added a section (SM G.10) where we illustrate the fact that both weight-importance methods and the hypernetwork approach can benefit from forward transfer of knowledge when sequentially learning tasks, while highlighting that the mechanisms for doing so differ between the two approaches.. Finally, we added a concurrent study to our related work section which addresses the complementary question of how to learn a set of weights that optimally allocates subspaces across tasks to allow transfer while preventing forgetting (Duncker et al. 2020, NeurIPS).

---

### Official Review · AnonReviewer1 · 2020-10-28
**An interesting and timely analysis of CL for RNNs**

**Rating:** 7
**Confidence:** 4

**Review:**

Summary:

The authors do an evaluation of the application of weight-importance continual learning methods to recurrent neural networks (RNNs). They draw out the tradeoff between complexity of precessing and just remembering (working memory) in terms of the applicability of these weight importance methods. They also provide some theoretical interpretation based on stying linear RNNs.

Overall, I vote for accepting this paper because the work is well-motivated, thorough, and provides useful insights. My major concerns are listed below.

Strengths:

+ The paper is very well written, and the motivation, methods and inferences are quite clearly described. The main question the authors are considering is very clear.
+ The results around use of existing continual learning methods to RNNs is timely and relevant.
+ The insight into the tradeoff between complexity of processing and working memory requirements and its effect on the ability of the network to continually learn is very interesting. Similarly the fact that hypernetwork based approaches work better than other approaches most of the time is useful.
+ The analysis of the above tradeoff using a linear RNN is also interesting since it provides a nice intuition for why the tradeoff exists.

Weaknesses:

- The motivation and conclusions from the ssMNIST task is not very evident and the tasks doesn't seem to make a clear point.
- Readability of some parts of the paper depend very heavily on the supplement, and the paper itself doesn't stand by itself. For example, the linear RNN analysis, description of some tasks (I had to read the supplement to actually understand the permuted copy and the pattern manipulation tasks). As an aside, I hope the authors submit an extended version to an appropriate venue, because I think many of the results and discussions relegated to the supplement seem quite interesting and relevant for the community (e.g. the task-conditional processing).
- An analysis of why hypernetworks perform better would be interesting. So would have been some proposals for methods designed specifically for CL with RNNs.

Minor:

The main text doesn't clearly mention that vanilla RNNs are used in section 5.1

---

> ### Author Response · Authors · 2020-11-20
> **Reply to AnonReviewer1 (1/2)**
>
> We thank you for your encouraging assessment and the valuable feedback. Below we describe the individual changes concerning your remarks point-by-point.
>
>   - *The motivation and conclusions from the ssMNIST task is not very evident and the tasks doesn't seem to make a clear point.*
>
> In brief, this benchmark allows us to investigate the effect of increasing working memory requirements in a real-world dataset. Indeed, the amount of information to be stored and manipulated per sample can be directly controlled by the number of digits per input sequence. Furthermore, in contrast to the Copy Task variants (which were designed to closely match the theoretical assumptions of the linear RNN analysis), this dataset allows investigating a scenario where task identity can be inferred from the input alone. Intriguingly, our results show that weight-importance methods are disproportionately affected by task complexity (compared to, for instance, the HNET approach). This shows that, although weight-importance methods could in theory perform task-conditional computation, in practice they cannot leverage this information efficiently. In the revised manuscript, we have rephrased the SSMNIST section such that the reason for integrating these experiments becomes more clear.
>
>   - *Readability of some parts of the paper depend very heavily on the supplement, and the paper itself doesn't stand by itself. For example, the linear RNN analysis, description of some tasks (I had to read the supplement to actually understand the permuted copy and the pattern manipulation tasks). As an aside, I hope the authors submit an extended version to an appropriate venue, because I think many of the results and discussions relegated to the supplement seem quite interesting and relevant for the community (e.g. the task-conditional processing).*
>
> We found AnonReviewer1’s enthusiasm regarding the supplementary insights and discussions encouraging and take the comment on readability very seriously. To improve readability, we expanded the paragraph that summarizes the analysis on linear RNNs (Sec. 4), such that the most important aspects necessary for understanding the analysis are covered. Furthermore, we made the description of the Copy Task variants more accessible i) by rephrasing the description such that these can be understood without having to refer to the SM, ii) by adding a schematic that illustrates the different Copy Task variants in the main text (Fig. 2), and iii) by elaborating on the schematic in the SM (Fig. S1) with a complete description of the inputs and outputs corresponding to each of the variants. Overall, these changes have made the main text more self-contained, while interesting pointers to further details and control experiments which we do not deem necessary for the main story of the paper remain in the SM.

---

> > ### Author Response · Authors · 2020-11-20
> > **Reply to AnonReviewer1 (2/2)**
> >
> >   - *An analysis of why hypernetworks perform better would be interesting. So would have been some proposals for methods designed specifically for CL with RNNs.*
> >
> > We thank the reviewer for this valuable comment. In the revised manuscript we report new complementary analyses that address these concerns.
> >
> > **Elaborating on HNET’s performance.** To thoroughly discuss why the HNET method performs better than other regularization approaches we added Sec. G.11 to the SM. There, we discuss in more detail why we consider HNET a suitable approach for CL in RNNs, and more generally, why it is an intriguing approach for CL. In addition, we more explicitly stress the fundamental differences with respect to weight-importance methods and, inspired by the reviewers’ request, examine solutions obtained with each of the two approaches. For this, we quantify the viability of the solutions and find that the HNET approach has the ability to find solutions in flatter regions of the loss landscape. This is very relevant in a CL setting, because such solutions, that tend to generalize better, will be more robust to perturbations introduced when learning new tasks. In turn, lower levels of regularization will be required to maintain previous knowledge, leading to increased plasticity for learning new tasks. Nevertheless, our empirical analysis reveals that the current HNET approach does not necessarily succeed in finding such solutions, and therefore opens interesting avenues for actively guiding towards flat minima in future work.
> >
> > **Elaborating on methods designed specifically for CL in RNNs.** While we do not propose new CL methods, our paper will guide future work in this area by providing a first thorough investigation of the strengths and weaknesses of established CL methods applied to RNNs. Besides fairly acquired baselines, we also provide key insights that can direct the development of CL methods tailored for RNNs. We summarize two major discussion points below.
> >
> >   - While this work focuses solely on CL methods that operate on RNNs with a static set of weights, it is an intriguing question whether more challenging practical scenarios can benefit from time-dependent processing (i.e. through the use of a different set of weights per timestep). As suggested by prior work, recurrent hypernetworks allow time-dependent processing in RNNs via time-specific weights. It is therefore a valuable insight to realize that the HNET approach can just as readily be applied to this setting by protecting the static set of weights maintained within the recurrent hypernetwork. Because the CL regularizer is agnostic to the more challenging architectural setup, one can expect it to perform similarly well.
> >
> >   - As you pointed out, we mention in our discussion that making the recurrent processing of weight-importance methods task-conditioned (e.g. by providing task identity as an additional input) could vastly improve their performance. The reason behind this is that the need to solve all tasks simultaneously within the hidden space would be overcome, and the amount of working memory that could be allocated to each task could be increased.
> >
> > We hope that this type of insight can inspire future work and that our provided baselines and code will facilitate the development of more tailored CL approaches for RNNs.
> >
> >   - *The main text doesn’t clearly mention that vanilla RNNs are used in section 5.1*
> >
> > Thank you for pointing this out; the text now specifically mentions that vanilla RNNs are used in Sec. 5.1.

---

### Author Response · Authors · 2020-11-20
**General response to all reviewers**

We thank the reviewers for the evaluation of our manuscript and for the overall positive assessment. We are grateful for the constructive feedback, which allowed us to substantially improve the paper. Below we briefly list how we addressed the main concerns and later provide further details in a point-by-point reply.

**First,** to address one of the main concerns raised by AnonReviewer4, we now show that our results from the theoretical analysis and the Copy Task generalize to real-world datasets and other RNN architectures. Specifically, to improve the relationship between these results and real-world problems, we conducted an additional SSMNIST experiment where, similarly to the Padded Copy Task, sequence lengths increase while working memory requirements remain fixed. Consistent with the Copy Task results, the performance of weight-importance methods is not affected by a mere increase in weight reuse (results in SM G.8). Encouraged by AnonReviewer4, we also investigated whether our Copy Task results generalize to LSTMs. For this, we repeated the Copy Task analyses and confirmed that the observed trends hold for this type of network.

**Second,** to address an important comment from AnonReviewer1, we provide a more detailed discussion and have performed additional analyses to gain novel insights into the factors leading to the superior performance of the hypernetwork approach as compared to weight-importance methods (SM G.11). Also, as suggested by AnonReviewer2, we now explicitly consider the case of task similarity and forward transfer (results in SM G.10). For this, we performed additional experiments that illustrate that both weight-importance methods and the hypernetwork approach can benefit from knowledge transfer when tasks are similar.

**Third,** as suggested by AnonReviewer1, we took a serious effort to improve the readability of the main text by elaborating explicitly on content that was initially relegated to the supplementary material. These improvements include, for example, adding a small figure with typical Copy Task patterns, which will make it easier for readers to understand the different variants of this dataset (Fig. 2). Furthermore, we extended the paragraph discussing the analysis of linear RNNs (Sec. 4), such that the logic and intuition can be understood without having to refer to the SM.


Given the new experiments and better readability, which considerably improved the manuscript, we kindly welcome all reviewers to reassess our paper and to re-evaluate their rating if they agree with the improvements. We are open to any suggestions that may further improve our manuscript and remain available to answer any question to support the reviewers in their assessment process.

---

### Decision · Program_Chairs · 2021-01-07
**Final Decision**

**Decision:**

Accept (Poster)

**Comment:**

I agree with the reviewers, and I find the careful analysis of CL approaches relying on regularization for RNN useful and insightful. I do feel that a lot of the interesting content is still in the appendix (from a quick skim and looking at the plots in the appendix) but I think something like this can potentially be unavoidable.

I do like the separation between sequence length and memory requirements. I think making observations about different types of recurrent architectures is hard, but I think the paper does a good job to raise some interesting questions.

A note that I would make (that I haven't seen raised through a quick look in the paper) is that is not clear how the Fisher Information Matrix should be computed in case of a recurrent model (which is a problem in general). E.g. a typical thing is to compute it as for a feed-forward model (using the gradients coming from BPTT) which is feasible computationally, but actually that is problematic as you first sum gradients before taking their outer-product rather than summing the outer-products corresponding of the different terms in the gradient. I'm wondering if that plays a role here as well.

Overall I think the paper does careful analysis and ablation studies and raises some interesting observation of how one should approach CL algorithms for RNN models.

---

> ### Author Response · Authors · 2021-03-10
> **Reply to Program Chairs**
>
> We thank the chair for the very encouraging comments.
>
> To address the valid concern: EWC only requires the diagonal elements of the Fisher Information matrix, which is why we do not need to compute the full outer product. What is however important is the correct specification of the negative-log-likelihood (NLL), which has to take the sequential nature of the problem into account as described in SM eq. 6 - 8. For clarity, we use the empirical Fisher, which is computed by averaging the following term across the training dataset $\bigg( \frac{\partial \text{NLL}_n}{\partial \psi_i} \bigg)^2$, where $\text{NLL}_n$ is the NLL computed for each sample in the dataset separately.